# Prepartum Magnesium Butyrate Supplementation of Dairy Cows Improves Colostrum Yield, Calving Ease, Fertility, Early Lactation Performance and Neonatal Vitality

**DOI:** 10.3390/ani13081319

**Published:** 2023-04-12

**Authors:** Levente Kovács, Ferenc Pajor, Mikolt Bakony, Hedvig Fébel, Joan E. Edwards

**Affiliations:** 1Institute of Animal Sciences, Hungarian University of Agriculture and Life Sciences, 7400 Kaposvár, Hungary; kovacs.levente@uni-mate.hu (L.K.); pajor.ferenc@uni-mate.hu (F.P.); 2Bona Adventure Ltd., 2100 Gödöllő, Hungary; 3Department of Animal Hygiene, Herd Health and Mobile Clinic, University of Veterinary Medicine, 1078 Budapest, Hungary; mbakony@yahoo.co.uk; 4Nutrition Physiology Research Group, Institute of Physiology and Nutrition, Kaposvár Campus, Hungarian University of Agriculture and Life Sciences, 2053 Herceghalom, Hungary; 5Palital Feed Additives B.V., 5334 LH Velddriel, The Netherlands; j.edwards@palital.com

**Keywords:** magnesium butyrate, dairy cows, colostrum, fertility, retained fetal membranes, milk yield, calf

## Abstract

**Simple Summary:**

Butyrate is an established feed additive for supporting gut health in pigs and poultry and also young ruminants where it is used to promote rumen development. However, the effect of supplementing dairy cows with butyrate ahead of calving has been less extensively studied, and it is not known what effect precalving butyrate supplementation has on the calf. The purpose of this on-farm study was to investigate the effect of magnesium butyrate (MgB) fed daily for the last 3 weeks before the expected time of calving on dairy cow and calf-related parameters. The study findings showed that prepartum MgB enhanced colostrum yield and the total yield of immunglobulin G, protein and lactose. Improved early lactation performance, calving ease, fertility and body condition score were observed in the cows of the MgB-supplemented group, and their calves had increased vitality at birth. These findings highlight that prepartum MgB supplementation of dairy cows gives a wide range of benefits, which also extend to the calf.

**Abstract:**

Butyrate promotes rumen epithelium growth and function; however, the effect of prepartum butyrate supplementation on dairy cow productivity, health and their offspring has not been extensively studied. Furthermore, no studies have investigated the effect of magnesium butyrate (MgB), which is also a source of magnesium. A trial was performed to test the hypothesis that prepartum MgB supplementation (105 g/cow/day) would increase colostrum quality and improve calving, newborn calf vitality and cow health. Multiparous Holstein cows were randomly assigned to MgB supplemented (*n* = 107) and Control groups (*n* = 112). Colostrum yield and the total yield of IgG, protein and lactose were higher (*p* ≤ 0.05) in the supplemented group. The calving assistance rate was lower (*p* ≤ 0.012), and the neonatal vitality score was higher (*p* ≤ 0.001) in the MgB group. Improved parameters related to cow health and fertility were observed in the supplemented group. The MgB group also had higher milk yield during the first week of lactation (*p* ≤ 0.001), and a higher (*p* ≤ 0.05) body condition score from 3 to 9 weeks after calving. In conclusion, prepartum MgB supplementation provides a wide range of benefits for dairy cows, as well as their newborn calves.

## 1. Introduction

Rumen development is stimulated by volatile fatty acids (VFA) which increase cell proliferation in rumen wall tissue [1]. Unlike acetate and propionate, butyrate is unique in also being an inhibitor of cell apoptosis in rumen wall tissue and has been shown to increase ruminal VFA transport proteins and rumen epithelial blood flow [2,3,4].

Building on this and the known benefits of butyrate for rumen development in young calves [1,5], Kowalski et al. [6] investigated the effect of sodium butyrate supplementation on mature ruminants. Feeding Holstein-Friesian bulls sodium butyrate resulted in increased rumen papillae length, rumen papillae thickness and rumen wall surface area.

As rumen wall surface area decreases during the dry period of cows by more than 50% [7], the effect of prepartum butyrate supplementation on dairy cows has also been studied. Kowalski et al. [6] found no benefit of butyrate on lactation performance when sodium butyrate was given to dairy cows prepartum. Recently, a study with calcium butyrate confirmed these findings [8].

Dietary strategy may be an important aspect to consider when testing the effect of prepartum butyrate supplementation on dairy cows [9]. For example, the rate of concentrate build-up during transition can affect the time needed to reach full rumen redevelopment [2].

Relative to calcium management strategies of transition cows, prepartum feeding of butyrate in the form of calcium or sodium salts also needs consideration. For example, calcium butyrate would not be advisable for cows fed low calcium content prepartum rations, and sodium butyrate has implications relative to the use of dietary cation-anion difference (DCAD). Magnesium butyrate (MgB) could be an interesting alternative, particularly as it would safeguard against the risk of magnesium deficiency adversely affecting calcium mobilization rates [10,11]. This could help reduce the risk of milk fever during early lactation and improve muscular contractions at partum. The latter is important as calving ease can influence the vitality of the newborn calf and the start of lactation, and this can have significant economic [12,13] and animal welfare outcomes [14,15].

Colostrum is also key to giving newborn calves the best start in life. In colostrum, immunoglobulins make up 70–80% of the total protein [16], and this is of particular importance as the transfer of passive immunity to the calf occurs through the colostrum and not via the placenta [17]. Relative to parturition, butyrate supplementation has been previously demonstrated to improve colostrum composition in pigs [18,19]. This was not confirmed in a dairy cow study by Hiltz et al. [20]. However, the authors indicated that the lack of a standardized colostrum collection time following calving may have prevented a significant treatment effect from being observed.

Based on the above literature, it was hypothesized that prepartum supplementation of MgB can increase colostrum quality and positively influence factors linked to calving, newborn calf vitality and cow health. In order to test this hypothesis, multiparous dairy cows with or without prepartum (last 3 weeks before the expected time of calving) MgB supplementation were studied. At partum, cows were monitored in terms of calving ease, colostrum quality and quantity. Newborn calves were examined at birth in terms of vitality, and monitored until weaning in terms of health. As the effect of prepartum MgB supplementation on postpartum performance and health has not been previously assessed in dairy cows, milk yield, body condition score (BCS), health disorders and fertility were also studied during the first 70 days in milk (DIM).

## 2. Materials and Methods

### 2.1. Ethical Statement

All methods and the applied procedures on the animals were performed in accordance with the relevant guidelines and regulations of the Pest County Government Office, Department of Animal Health (Permit Number: PE/EA/1973-6/2016) that approved the study.

### 2.2. Animals, Housing and Experimental Design

The experiment was carried out on a large-scale dairy farm in Hungary (47°18′191″ N 18°48′336″ E), which has a herd of over 900 lactating Holstein Friesian cows. The farm was visited for a 9-month period between January and September 2021. A matched pair design was applied to control for confounders and to balance control and experimental groups for previous lactational milk yield, parity, body condition and time of expected calving. Two hundred-and-thirty-two multiparous Holstein-Friesian cows that were expected to calve between February and June were enrolled in the study. Forty days before the start of the trial, cows were blocked by the expected calving date and randomly assigned to receive diets containing MgB supplementation (MgB, *n* = 116) or the regular diet with no supplementation (Control, *n* = 116) in the prepartum period.

Before 23 days of the expected time of calving until parturition, 70% MgB encapsulated in a fat matrix (Rumen-Ready^®^, Palital Feed Additives B.V., Velddriel, The Netherlands) was added to the prepartum total mixed ration (TMR) of MgB cows at a dosage level of 150 g/cow/day (i.e., 105 g of MgB/cow/day). Rumen-Ready^®^ is formulated so that the MgB is fully released in the rumen in a gradual manner. The MgB supplement was mixed into the prepartum TMR before morning feeding according to the number of cows in the MgB group. The prepartum TMR was pushed in towards the pen eight times daily. The amount of feed refusal was checked before evening feeding, in order to monitor MgB intake, and no feed refusals were observed.

Directly before enrollment, each animal was examined clinically and cows with a suboptimal BCS (<3 or >4.25), locomotion scores higher than 2 [21] or exhibiting any sign of disease were excluded. Animals with less than 18 days or more than 25 days of prepartum TMR with or without MgB supplementation were excluded from the dataset. Finally, 112 Control [means ± SD; parity = 3.03 ± 0.13 (parity 2; *n* = 45, parity ≥ 3; *n* = 67), previous lactational milk yield = 9681 ± 143, BCS = 3.61 ± 0.02] and 107 MgB cows participated in the trial [means ± SD; parity = 2.94 ± 0.08 (parity 2; *n* = 41, parity ≥ 3; *n* = 66), previous lactational milk yield = 9736 ± 133, BCS = 3.61 ± 0.02]. At the start of the trial, 10 Control cows had a locomotion score of 2, and 102 Control cows had a locomotion of score of 1, whereas 8 MgB cows had a locomotion score of 2, and 99 MgB cows had a locomotion score of 1 at the trial start.

The monthly average temperature humidity index (THI), the number of Control and MgB animals enrolled monthly in the trial, and changes in average and maximal daily THI in the barns of Control and MgB cows are presented in Appendix A. Further details of the number of cows enrolled in the study according to parity and experimental group are provided in Appendix A.

From 28 d before expected calving, dry cows were housed in a prepartum pen (measuring 35 m × 20 m), which included 50–60 animals and was bedded with deep straw. Before calving, cows were fed a prepartum TMR ad libitum containing a dietary forage-to-concentrate ratio of 81.7:18.3 on a dry matter (DM) basis. The prepartum TMR was fed twice daily (early morning and late evening). Calvings took place in the prepartum pen or, if continuous supervision or obstetrical assistance was required at calving, in a separate maternity pen. Assistance by well-trained farm personnel was performed at the latest within 90 min after the appearance of the amniotic sac in the vulva as described previously by Kovács et al. [22]. Newborn calves were removed from the dams immediately after birth. Within 1 h after calving the dam was milked, and 3.8 L fresh colostrum was provided to the newborn calves by nipple bottle.

During the first 5 DIM, cows were housed in postpartum pens, each including four animals, and were milked twice daily at 0400 and 1400 h in a four-stall herringbone milking parlor operated with DeLaval Control Valve bucket milking machines (DeLaval International AB, Tumba, Sweden). After 5 DIM, cows were introduced to the fresh lactation group and milked twice daily at 0500 and 1500 h in a 2 × 28-stall parallel Bosmark milking parlor (Bosmark Kft., Biatorbágy, Hungary). Cows were fed a postpartum TMR twice daily with a 57.7:42.3 forage to concentrate ratio on a DM basis until 90 DIM and water was available ad libitum. The composition of prepartum and postpartum TMRs did not change during the study period. The ingredient and chemical composition of prepartum and postpartum TMRs are shown in Table 1. The chemical composition of the prepartum and postpartum TMRs was analyzed with a 4-week sampling frequency from the start of MgB-feeding until 70 DIM of the last cow enrolled in the study using standardized laboratory methods [23,24].

### 2.3. Colostrum Quantity and Quality

Within 1 h after calving, all four udder quarters of the dams were completely milked by a portable milking machine into a steel churn. Colostrum was then transferred into plastic graded milk buckets and the volume (L) was recorded (hereinafter referred to as colostrum yield). Colostrum from each cow was agitated for at least 2 min, to ensure an even distribution of constituents, before a 50 mL sample was placed into a large plastic vial (50 mL). This vial was used for fat, protein, lactose, and total solids contents analysis. At the time of collecting samples for the colostrum chemical composition analysis, a 2 mL aliquot was also collected in a 2 mL plastic vial for colostral IgG measurement. All samples were frozen at −20 °C until analysis. Frozen colostrum samples (50 mL) were thawed in a water bath at 40 ± 2 °C, and vials were inverted 10 times to thoroughly mix the colostrum and secure an even distribution of constituents. Colostrum fat, protein, lactose and solids non-fat concentrations (%) were determined using the LactoScope infrared spectrometry analyzer (Delta Instruments, Drachten, The Netherlands). It should be noted that the colostrum samples met the microbiological quality requirements for bacterial contamination (≤100,000 CFU/mL). Colostrum IgG concentrations (mg/mL) were determined using a Bovine IgG Radial Immunodiffusion Test Kit which had a sensitivity range of 200 to 3000 mg/dL (#728411, Triple J Farms, Bellingham, WA, USA). Manufacturer’s instructions were followed when using the kit with the exception that colostrum samples were first diluted in water before being analyzed.

### 2.4. Calving Ease

Calving ease (i.e., the incidence and degree of dystocia) was assessed by the first author based on the recordings of a closed-circuit camera system including two day/night outdoor network bullet cameras (Vivotek IP8331, VIVOTEK Inc., New Taipei City, Taiwan) installed above the prepartum group pen allowing the identification of individual cows. In cases of assisted calvings, two portable video cameras (Legria HF M36, CANON, Tokyo, Japan) were used after placing the cows into the maternity pen. The incidence and degree of dystocia was scored using a 4-point scale as follows: (1) eutocia (normal calving) was recorded as no assistance at calving [25]; (2) light dystocia was considered in cases of prolonged spontaneous calvings (>2 h from hooves appearance to delivery) and/or calvings assisted by one person without the use of mechanical traction, with moderate force; (3) mild dystocia required assistance by two people without mechanical traction, with considerable force; (4) severe dystocia was recorded after assistance by three people with considerable force or assistance with the use of mechanical extraction during delivery [22].

### 2.5. Calf Vitality and Health

A total of 232 calves (*n* = 116 born to Control dams, and *n* = 116 born to MgB dams) including twins (*n* = 10 Control, and *n* = 16 MgB) were examined in the study. Calf sex was determined immediately after birth. Calf vitality was assessed immediately after birth by a trained individual who was blinded to treatment groups. The calf vitality criteria recommended by Szenci [26] were used in a 12-point scoring system as follows: (1) muscle tone (2: normal, 1: low, 0: toneless); (2) erection of the head (2: erected head, 1: head requiring support; 0: head dropping); (3) muscle reflexes (2: normal reflectory movements, 1: reduced number and intensity of reflectory movements, 0: limbs extended); (4) mucus membrane color (2: pink, 1: pale pink, 0: cyanotic); (5) heart rate (2: normal/regular 120–220 bpm, 1: bradycardia/irregular <120 bpm, 0: absent); (6) sucking drive (2: intensive, 1: reduced, 0: absent). Following the vitality assessment, scores were summed up. Calves with score 0 at birth and calves dead within 24 h of life were considered stillborn. Calf birth weight was measured within 1 h after birth using a calibrated digital weigh scale. Calf hip height and hip width were measured in cm using a meter stick within 30 min after delivery.

Live calves with adequate colostrum uptake (3.8 L) from their own dams (Control; *n* = 108; MgB, *n* = 105) were visually inspected by a trained investigator (who was blinded to treatments) from birth until weaning (50 d of life) three times per week (i.e., on a Monday-Wednesday-Friday basis). A systematic scoring system developed for the assessment of bovine respiratory disease (BRD) in pre-weaned dairy calves was used [27] to assess six clinical signs as follows: Cough = 2 points, Eye discharge = 2 points, Fever (≥39.2 °C) = 2 points, Abnormal respiration = 2 points, Nasal discharge = 4 points, Ear droop or head tilt = 5 points. A total score of five points or higher characterized a BRD case. Neonatal calf diarrhea (NCD) was diagnosed according to the Calf Health Scoring Criteria by the University of Wisconsin-Madison [28]. The calf mortality rate was also calculated for both groups based on the period between 24 h and 50 d of life.

### 2.6. Fertility and Health of Dairy Cows

Blinded to the treatment groups, inspections for the determination of health status were carried out by a trained veterinarian during the entire study period using generally accepted clinical diagnostic methods and diagnoses, and treatments (if necessary) were recorded. Health disorders diagnosed during the trial included milk fever, puerperal metritis, abomasum displacement, mastitis, clinical ketosis, and other digestive disorders. For each individual cow, examinations were conducted daily during the first 5 DIM, and then every three days throughout the remainder of the first 70 DIM. Retained fetal membranes (RFM) were defined as the non-expulsion of fetal membranes beyond 24 h after calving [29,30], and each cow was examined for RFM at 24 h after calving. The interval to first heat was calculated based on the data recorded by the HeaTime Ruminact System (SCR Engineers Ltd., Netanya, Israel) which is validated for heat detection in dairy cows. Heat alerts were confirmed by clinical examinations. The number of services per conception (SPC) and the interval to conception from calving without later embryonic/early fetal mortality were recorded based on the diagnoses of on-farm routine veterinary inspections (i.e., transrectal ultrasonography and transrectal palpation confirmation). Embryonic/early fetal mortality was determined by means of two transrectal ultrasonographic examinations, the first between 27 and 30 d after artificial insemination (AI) and the second at 45 d after AI, respectively.

### 2.7. Milk Production, Body Condition Score and Lameness Status of the Cows

Daily milk yield (kg) was recorded during the first 70 DIM using on-farm records from the milking system. The BCS of the cows was scored weekly from 255 to 260 d of pregnancy until 70 DIM (i.e., −3, −2, −1, 0, and 1 to 10 weeks after calving) using the 5-point USA scoring system [31] with 0.25-point increments. Cows were scored by both looking at and handling the backbone, loin and rump areas. Since the pin bone, hip bone, top of the backbone and ends of the short ribs do not have muscle tissue covering them, any covering you see or feel is a combination of skin and fat deposits. BCS range from 1 (a very thin cow with no fat reserves) to 5 (a severely over-conditioned cow). Ideal condition scores are 3.0–3.25 at dry-off and calving, and 2.25–2.75 at peak lactation. Locomotion scorings were carried out immediately after BCS scoring on a 5-point scale according to Sprecher et al. [21]. Cows were observed walking on flat, nonslip concrete in a well-lit location [32]. The BCS and locomotion scoring were determined by the same trained person who was blinded to the groups.

### 2.8. Statistical Analyses

Statistical analyses were performed in the R–4.1.2 statistical environment and language [33]. The level of significance was in all tests set at *p* < 0.05. Data were tested for constant variance (Levene’s test) and the Shapiro–Wilk test was used for testing the normality of data. Means of colostrum-related variables were compared between groups using a two-way ANOVA with treatment (Control vs. MgB), level of parity (parity = 2 vs. parity ≥ 3) and their interaction as factor variables. As well as the colostral protein, fat, lactose and solids non-fat concentrations (g/kg), the total amount of these components was calculated (g) and compared between groups.

A calving ease score above 1 was classed as assisted calving. The relative frequency of assisted calvings was compared between groups using Fisher’s exact test. The means of calf birth weight, hip height, hip width and vitality score were compared between calves born to Control and MgB dams using Welch’s *t*-test. Late embryonic/early fetal mortality rate, stillbirth rate, the prevalence of NCD and BRD, and the preweaning mortality rates were compared between groups by Fisher’s exact test.

The SPC was compared between treatment groups separately for different levels of parity by means of the Wilcoxon–Mann–Whitney test. The interval from calving to first heat and calving to conception was considered time-to-event data, which included censored observations due to culling before first heat/conception or unobserved outcome. Median days from calving to first heat and calving to conception for different levels of parity are presented as Kaplan–Meier estimates. Medians of intervals between treatment groups adjusting for the level of parity were compared by means of the stratified log-rank test with the treatment group as the main effect and the level of parity as the stratum. The relative frequency of RFM cases, other health disorders (i.e., milk fever, puerperal metritis, abomasum displacement, mastitis, clinical ketosis, and other digestive disorders) and culling rate before 70 DIM were compared between Control and MgB groups using the Fisher’s exact test. The distribution of the number of animals according to the lameness categories was compared between groups separately for each sampling week using the Fisher’s exact test. *p*-values were then adjusted for multiple comparisons, by controlling for false discovery rate.

Daily milk yields per group were averaged over weeks. Weekly average milk yields were then compared between groups (Control and MgB) and weeks by fitting a linear mixed-effects model. The response variable was the daily average milk yield and fixed effects were treatment, week as a categorical variable and their interaction term. To take into account the correlation of repeated measurements on the same animal, cow identity (ID) was included as a random term that enables separating individual variation and main effects of interest. The level of significance for fixed effects was set at *p* < 0.05. Differences in mean average milk yield between Control and MgB groups were compared by posthoc single-step multiple comparisons based on studentized range distributions conditioning for week controlling for the family-wise error rate using Tukey’s *p*-value adjustment.

Mean BCS was compared across groups and weeks by fitting a linear mixed effects model, similar to milk yield results. The fixed effects were treatment, week, and their interaction. Cow ID was included in the model as a random effect. Differences in mean BCS between Control and MgB groups each week were compared by posthoc tests as described for milk yield. The level of significance for main effects and all comparisons was set at *p* < 0.05.

## 3. Results

The MgB cows spent 23.2 ± 1.4 d (range; 18–24 d) in the prepartum pen receiving MgB supplementation before calving, whereas Control cows spent 23.0 ± 1.3 d (range; 19–25 d) in the prepartum pen before calving.

### 3.1. Colostrum Quality and Quantity

Colostrum yield and chemical analysis variables of Control and MgB cows are presented in Table 2. Colostrum yield was higher in the MgB group compared to the Control group. Except for total fat, the yield of all colostrum-related variables (total IgG, total protein, total lactose, total solids non-fat) were greater in MgB cows compared to those produced by the Control group. There was no difference in the concentrations of colostrum-related parameters between groups. Parity affected neither colostrum yield nor parameters related to colostrum quality. No group × parity interactions were found in terms of colostrum yield and quality.

### 3.2. Newborn Calf Vitality, Size and Health

The vitality scored immediately after birth was significantly higher (*p* < 0.001) in newborn calves born to MgB dams compared to calves born to Control dams (Table 3).

Neither birth weight nor hip width or hip height differed between calves born to Control and MgB dams (Table 3).

The mean colostral IgG amount fed to Control and MgB calves was calculated to be 271.7 ± 23.5 g and 269.4 ± 21.8 g, respectively. None of the parameters related to the health of the calves were significantly influenced by the treatment. Stillbirth (5.2% vs. 5.2%, *p* = 1.000) and mortality rates (2.6% vs. 1.7%, *p* = 1.000) were similar between Control and MgB groups, respectively, and the prevalence of NCD (19.0% vs. 11.2%, *p* = 0.141) and BRD (12.1% vs. 5.2%, *p* = 0.099) showed also no significant difference between groups. Data described in this section are not shown elsewhere.

### 3.3. Cow Fertility and Health

The median number of days to first heat, the median number of SPC and the median number of days from calving to conception were significantly lower in the cows of the MgB group compared to the Control group (Table 4) for both levels of parity.

The health-related parameters of cows in the Control and MgB groups diagnosed within the first 70 DIM are presented in Table 5. The overall incidence of calving assistance was 58.2%, and cows in the Control group needed significantly more assistance than those in the MgB group. The prevalence of RFM cases, as well as embryonic/early fetal mortality rates, were significantly lower for cows in the MgB group compared to the Control group. There was no significant difference between the groups in terms of the prevalence of health disorders or in the culling rate within 70 DIM. The lameness status of cows was similar across groups with no significant differences, including in the distribution of lameness scores between Control and MgB cows during the experiment (Appendix A).

### 3.4. Milk Yield and BCS

Changes in weekly averages of daily milk yield are shown in Figure 1 for cows from the Control and MgB groups.

Increases in milk yield showed a similar pattern in both groups with a group × week interaction (*p* < 0.001). The weekly average of daily milk yield was higher in cows from the MgB group during the first week postpartum (*p* < 0.001) compared to the Control group. There were no significant differences between groups in average daily milk production for the other weeks of lactation until 70 DIM. Mean BCS was similar across groups at 3, 2 and 1 weeks prior to calving and at parturition (Figure 2). After parturition, a gradual decrease in BCS was observed in both groups, which was moderate in MgB cows compared to Control cows with a group × week interaction (*p* < 0.001). Between weeks 3–9 of lactation, mean BCS was higher for cows in the MgB group compared to the Control group.

Further details of the milk yield and BCS are provided in Appendix A.

## 4. Discussion

The main objectives of the present study were to: (a) test the hypothesis that prepartum supplementation of MgB would improve colostrum quality and positively influence factors linked to calving, newborn calf vitality and health, and (b) assess the subsequent effect of prepartum MgB supplementation on postpartum milk yield, BCS, health disorders and fertility.

As the study was conducted on a commercial dairy farm, the findings can be considered to be reflective of what may happen in practice when MgB is supplemented prepartum. However, as a consequence, there are certain limitations that need to be considered when interpreting the study findings. Dietary supplementation of MgB was not able to be conducted at an individual animal level, but only at a group level. This meant that the exact amount of MgB consumed by each animal could not be determined. However, when the prepartum TMR was supplemented with MgB no feed refusals were observed. This indicates that, on average, the intended dosage of 105 g MgB/cow/day was achieved. The inability to determine feed intake on the farm is also a limitation of the study, and this point is discussed elsewhere in this section.

### 4.1. Colostrum Quality and Quantity

To the best of our knowledge, this is the first study showing the benefit of MgB on colostrum yield. Previous pig studies with sodium butyrate have not assessed this parameter, only colostrum composition, presumably due to practical issues with measuring colostrum yield [18,19,34]. However, it is interesting to note that sodium butyrate supplementation of sows resulted in a higher piglet body weight at 7 days of age which was not evident at birth [35]. Recently, it has been reported that increasing metabolizable energy during the late gestation of beef cattle can increase colostrum yield [36]. As the difference in the chemical composition of the prepartum diets in this study was minimal, this seems to be an unlikely explanation for the increased colostrum yield observed in this study, unless prepartum feed intake was increased. Due to practical constraints, the feed intake of individual cows was not able to be determined during this study either pre- or postpartum. However, Kowalski et al. [6] reported that prepartum supplementation of sodium butyrate did not affect DMI prior to calving. As such, it is suggested that butyrate supplementation may have improved colostrum yield via increasing ruminal VFA absorption, consistent with the known positive effects of butyrate on ruminal VFA transport proteins, tissue development and epithelial blood flow [2,3,4,6].

Bovine colostrum quality is associated with the most abundant immunoglobulin, IgG [37], and has been positively linked to lactation performance in later life [38,39]. Magnesium butyrate supplementation did not affect colostrum IgG concentration in this study, although it should be noted that the colostrum of all the cows was >50 g/L IgG which is considered to be an indication of high-quality colostrum [40]. Several authors have observed improved colostrum quality with increasing numbers of lactations [41,42,43,44,45]; however, consistent with our findings other studies have also reported no correlation between parity and colostrum IgG concentration [46,47,48].

A negative correlation exists between colostrum yield and IgG concentration [49,50], including when colostrum yield is improved by increasing late gestation metabolizable energy [36]. This was not the case with MgB supplementation, with the concentration of all colostrum components remaining the same due to their total amount increasing in line with the increase in colostrum yield. This indicates that MgB supplementation can increase colostrum yield without compromising its composition, which is an improvement that is in line with the known beneficial impact of butyrate on colostrum composition in pigs [18,19,35].

In earlier studies, experts have estimated a minimum mass of 100 g of IgG in the first colostrum feeding is needed to achieve successful passive transfer of immunity in an average 43 kg Holstein calf [51,52], while others [53] found that feeding 100 g of colostral IgG by esophageal intubation was insufficient to achieve this target. The latter authors concluded that at least 150 to 200 g of colostral IgG is required for successful passive transfer of immunity. In our study, the relatively low prevalence of NCD and BRD cases found in both groups was most likely linked to a large amount of colostral IgG fed to calves born to Control (271.7 g) and MgB dams (269.4 g). Although the prevalence of NCD and BRD were 7.8% and 6.9% lower in the calves born to MgB dams, compared to Control calves the differences did not reach significance. The moderately improved health condition of MgB calves indicates possible positive effects of MgB supplementation on other colostrum bioactive components [54], as colostrum IgG concentration and amount fed did not differ between the groups.

### 4.2. Newborn Calf Vitality and Health

Between 2 and 10% of all calves are born dead or die within 48 h after birth [55]. The stillbirth rate found here was comparable with previous studies in different countries with similar calving technologies [22,56,57]. No significant differences were found here between groups in terms of stillbirth and mortality rates; however, neonatal vitality was greater in calves born to MgB dams compared to Control dams. It is hypothesized that this was due to the improved calving ease of MgB dams rather than better in utero development, as no difference in calf weight or size at birth was found between the groups. The underlying basis for the improved calving ease is not clear but may be linked to improved stimulation of calcium mobilization for uterine contractions due to MgB being fed in addition to the prepartum TMR, as well as it being a highly bioavailable source of magnesium [58].

### 4.3. Cow Fertility, Health and BCS

The duration of retention of fetal membranes is highly dependent on uterine muscular contractions [59], and the incidence of RFM was 10.7% lower in the MgB group (3.7%) compared to the Control group (14.4%). Again, this may be linked to the provision of additional magnesium with the supplementation. However, other previously identified nutrition risk factors for RFM cannot be excluded [33,60], as increased rumen tissue development by butyrate may enhance the uptake of other micro- and macronutrients. Regardless of the exact mechanism, the ability of MgB to decrease the incidence of RFM is clearly beneficial as the average annual incidence of RFM varies from 3 to 10–12%, but is highly variable between farms and can reach up to 30–34% [61,62].

Cows with RFM have an increased risk of developing metritis, ketosis, mastitis, and even loss of their subsequent pregnancy [30]. Despite the decreased RFM rate in the MgB group, the culling rate within 70 DIM and the prevalence of postpartum health disorders and health disorders associated with increasing milk production were similar in the MgB and Control groups. Overall, the prevalence of health disorders was generally low in both groups, and the reason for this is likely (at least in part) to be due to the fact that animals with high disease risk were not included in the animals enrolled in the study.

A shorter interval to first heat, lower number of SPC and the interval to conception in MgB cows indicate a possible association between improved fertility and prepartum Mg-butyrate supplementation. The number of SPC reported in the present study is higher than those reported by others for Holstein-Friesian cows [63,64,65]. This is due to the systematic selection for milk production in Hungary (over 11,200 kg/305 d corrected lactation) and the handling of fertility-related parameters as secondary traits by Hungarian cattle breeders. It is also important to note that cows in the MgB group had better BCS, suggesting that their negative energy balance was not as deep as that of the Control group. Thus, we speculate that improved fertility was associated, at least in part, with the better body condition of the cows in the MgB group. The basis for the improved postpartum BCS is unclear, particularly as feed intake could not be determined in the study. As such, it remains to be determined if the improved postpartum BCS was associated with increased feed intake or improved feed efficiency.

### 4.4. Milk Yield

In the present study, cows with MgB supplementation produced more milk during the first week of lactation compared to the Control animals. This finding is in contrast to that of other studies, with either sodium butyrate [6] and calcium butyrate [8], which failed to find a significant benefit on lactation performance when supplemented to dairy cows prepartum. It is not clear to what extent the lower number of experimental animals (<20 per group) and/or type of diet in these studies may have contributed to these contrasting findings. It is unclear if the increased milk yield in this study was associated with a difference in feed intake, although prepartum butyrate supplementation has been reported previously to not affect pre or postpartum feed intake [6,8].

Another factor to consider is the type of butyrate supplementation used and its encapsulation. Kowalski et al. [6] used a sodium butyrate product that had a 70% triglyceride matrix coating and resulted in the butyrate releasing from the supplement within 16 h of ingestion. In the present study, MgB was used and it was encapsulated in a 30% triglyceride matrix (Rumen-Ready^®^) that enabled a much faster and complete release of the magnesium butyrate in the rumen (data not shown). As cows with RFM produce 9.79 L of milk/d less compared to their healthy counterparts [66], this may also have at least in part contributed to the observed improvement in lactation performance. Furthermore, as the BCS of MgB cows remained higher than that of Control cows from the third week of lactation without any significant impact on milk yield, it appears that MgB supplementation may either directly or indirectly influence nutrient and/or energy partitioning. It is interesting to note that high ruminal infusions of sodium butyrate (singly dosing either 1 or 2 g/kg body weight of butyrate) had been previously reported to influence insulin levels [67]. However, it should be noted that in this study the beneficial effects on milk yield and BCS occurred weeks after MgB supplementation had finished.

## 5. Conclusions

Based on the study findings, it can be concluded that prepartum MgB supplementation has a beneficial effect on calving ease, colostrum yield, milk production, BCS, and fertility. The vitality of newborn calves born to MgB dams was also better compared to calves from Control dams. In summary, due to its numerous positive effects, MgB can be advised to be fed to dairy cows prepartum. Further studies are needed to gain additional insight into the basis by which prepartum MgB supplementation benefits post-partum cows, including assessment of metabolic parameters and rumen fermentation.

## Figures and Tables

**Figure 1 animals-13-01319-f001:**
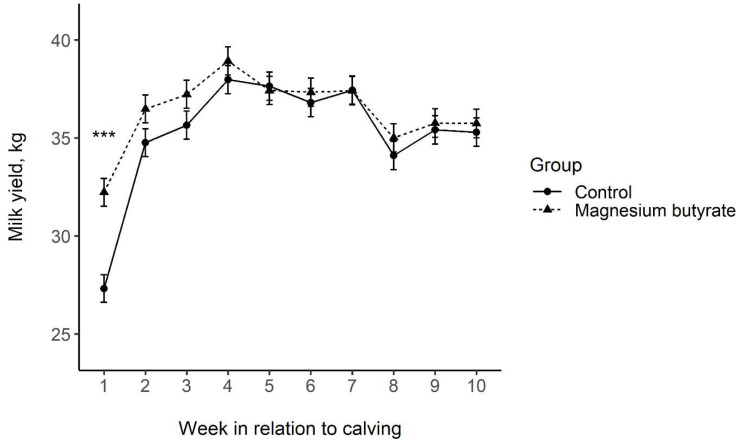
Weekly averages of daily milk yields of Control (●, *n* = 112) and Magnesium butyrate (▲, *n* = 107) groups during the first 70 DIM. Statistical significance is based on pairwise comparisons after fitting a linear mixed effects model. Results are presented as estimated marginal means (±SEM); *** *p* < 0.001.

**Figure 2 animals-13-01319-f002:**
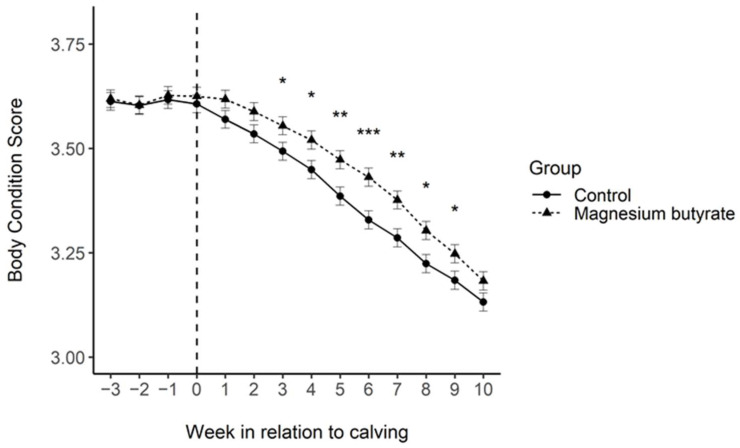
The evolution of BCS in cows of Control (●, *n* = 112) and Magnesium butyrate (▲, *n* = 107) groups from −3 to 10 weeks in relation to calving. Statistical significance is based on pairwise comparisons after fitting a linear mixed effects model. Results are presented as estimated marginal means (±SEM); * *p* < 0.05; ** *p* < 0.01; *** *p* < 0.001.

**Table 1 animals-13-01319-t001:** Ingredient and chemical composition of prepartum and postpartum total mixed rations for the Control and Magnesium butyrate groups.

Item		Prepartum		Postpartum ^2^
		Control	Magnesium Butyrate ^1^	
Ingredient, % DM				
	Corn silage	50.5	49.2	34.7
	Rye silage	-	-	6.24
	Alfalfa silage	8.90	8.67	7.45
	Grass hay	5.96	5.82	3.06
	Straw	10.4	10.1	-
	Brewer’s grains	5.96	5.82	6.24
	Corn flour	5.05	4.93	22.0
	Oats	2.52	2.46	5.75
	Extracted rapeseed meal	2.52	2.46	7.45
	Extracted sunflower meal	6.31	6.16	5.08
	Premix Mipro Pren 250 ^3^	1.88	1.84	-
	Premix Mipro RB 600 ^4^	-	-	2.03
	Rumen-Ready^® 1^	-	2.54	-
Chemical composition ^5^				
	Dry matter, g/kg	437	441	426
	NE_l_, MJ/kg of DM	6.23	6.32	7.25
	Ash, g/kg of DM	84.0	86.0	71.0
	Crude protein, g/kg of DM	114	114	167
	Ether extract, g/kg of DM	26.0	29.0	35.0
	Crude fiber, g/kg of DM	241	232	143
	NDF, g/kg of DM	507	494	339
	ADF, g/kg of DM	303	280	190
	ADL, g/kg of DM	50.0	47.0	39.0
	Starch, g/kg of DM	154	158	283
	Sugar, g/kg of DM	26.0	26.0	29.0

^1^ For the MgB group, the morning feeding of the prepartum TMR was supplemented with Rumen-Ready^®^ which is 70% MgB encapsulated in a fat matrix (Palital Feed Additives B.V., Velddriel, The Netherlands), and the prepartum TMR fed in the evening was the same as for the Control group. ^2^ Both groups received the same postpartum total mixed ration. ^3^ Contained Ca, 18.0%; P, 2.0%; Na, 7.5%; Mg, 8.0%; S, 5.1%; Zn, 4000 mg/kg; Mn, 3000 mg/kg; Cu, 900 mg/kg; I, 93 mg/kg; Se, 18 mg/kg; Co, 12 mg/kg; vitamin A, 300,000 IU/kg; vitamin D3, 100,000 IU/kg; vitamin E, 4000 mg/kg; vitamin B1, 24 mg/kg; vitamin B2, 12 mg/kg, vitamin B6, 6 mg/kg; vitamin B12, 60 μg/kg; niacin, 360 mg/kg; Ca-D-pantothenate, 24 mg/kg; choline chloride, 12,000 mg/kg; biotin, 15,000 μg/kg (Sano Ltd., Csém, Hungary). ^4^ Contained Ca, 16.0%; P, 0.8%; Na, 9%; Mg, 4.2%; S, 0.4%; Zn, 2500 mg/kg; Mn, 2500 mg/kg; Cu, 500 mg/kg; I, 75 mg/kg; Se, 15 mg/kg; Co, 10 mg/kg; vitamin A, 200,000 IU/kg; vitamin D3, 35,000 IU/kg; vitamin E, 2000 mg/kg; vitamin B1, 20 mg/kg; vitamin B2, 10 mg/kg; vitamin B6, 5 mg/kg; vitamin B12, 50 μg/kg; niacin, 300 mg/kg; Ca-D-pantothenate, 20 mg/kg; choline chloride, 48,000 mg/kg; biotin, 14,000 μg/kg; methionine, 0.85%; sugar, 6.7%; urea, 8.4% (Sano Ltd., Csém, Hungary). ^5^ Averaged values of the chemical composition analysis of prepartum and postpartum total mixed rations. Feed samples were collected monthly between February and October 2021.

**Table 2 animals-13-01319-t002:** Colostrum yield and chemical analysis variables (least squares means and standard error) of Control (*n* = 112) and Magnesium butyrate (*n* = 107) cows.

Item	Control		Magnesium Butyrate		*p*-Values ^1^		
	Parity = 2 (*n* = 45)	Parity ≥ 3 (*n* = 67)	Parity = 2 (*n* = 41)	Parity ≥ 3 (*n* = 66)	Group	Parity	Group × Parity
Colostrum yield (kg)	7.55 ± 0.65	8.23 ± 0.52	9.68 ± 0.66	9.48 ± 0.55	0.006	0.686	0.488
IgG concentration (mg/mL)	69.2 ± 4.08	74.3 ± 3.31	71.5 ± 4.13	69.1 ± 3.47	0.547	0.698	0.324
Total IgG (g)	535 ± 53.8	601 ± 43.2	675 ± 53.1	679 ± 45.4	0.034	0.473	0.524
Fat concentration (%)	6.10 ± 0.53	6.67 ± 0.43	5.47 ± 0.53	6.51 ± 0.46	0.496	0.097	0.662
Total fat (g)	456 ± 66.4	544 ± 54.7	576 ± 67.2	644 ± 57.5	0.076	0.206	0.877
Protein concentration (%)	17.0 ± 0.99	18.3 ± 0.82	17.8 ± 1.01	19.2 ± 0.86	0.363	0.160	0.975
Total protein (g)	1187 ± 120	1446 ± 99.2	1611 ± 122	1765 ± 104	0.001	0.065	0.641
Lactose concentration (%)	2.70 ± 0.12	2.50 ± 0.10	2.67 ± 0.12	2.58 ± 0.11	0.869	0.157	0.536
Total lactose (g)	215 ± 22.9	212 ± 18.9	267 ± 23.2	249 ± 19.9	0.041	0.621	0.735
Solids non-fat concentration (%)	24.9 ± 0.94	26.3 ± 0.74	25.1 ± 0.92	26.6 ± 0.78	0.790	0.083	0.998
Total solids non-fat (g)	1797 ± 164	2111 ± 135	2375 ± 166	2496 ± 142	0.002	0.152	0.533

^1^ Statistical significances between Control and Magnesium butyrate groups are based on results from the two-way ANOVA.

**Table 3 animals-13-01319-t003:** Birth related parameters of dairy calves born to Control and Magnesium butyrate dams (means ± standard deviations).

Item	Control (*n* = 116)	Magnesium Butyrate (*n* = 116)	*p*-Value ^1^
Vitality score ^2^	9.20 ± 2.57	10.6 ± 2.64	<0.001
Birth weight (kg)	43.6 ± 4.76	42.8 ± 4.89	0.259
Hip width (cm)	20.3 ± 1.80	20.0 ± 1.91	0.222
Hip height (cm)	77.8 ± 7.21	76.8 ± 7.31	0.327

^1^ Statistical significances between Control and Magnesium butyrate groups are based on results from the Welch’s *t*-test. ^2^ Neonatal vitality was scored using a 12-point linear scoring.

**Table 4 animals-13-01319-t004:** Median (and 95%CI for intervals, min-max for SPC) of parameters related to fertility of Control (*n* = 112) and Magnesium butyrate (*n* = 107) cows.

Item	Control	Magnesium Butyrate	*p*-Value ^1^
**Parity = 2**			
Interval to first heat (d postpartum)	23.0 (22; 25)	20.5 (19; 21)	0.01
Services per conception (SPC)	4 (2; 6)	2 (1; 6)	<0.0001
Interval to conception (d postpartum)	109 (106; 134)	77 (63; 99)	<0.0001
**Parity ≥ 3**			
Interval to first heat (d postpartum)	23 (22; 25)	20 (20; 22)	0.01
Services per conception (SPC)	4 (2; 6)	3 (2; 6)	0.0115
Interval to conception (d postpartum)	106 (94; 121)	84 (81; 94)	<0.0001

^1^ Statistical significances between Control and Magnesium butyrate groups are based on results from stratified log-rank test for intervals and Mann-Whitney-Wilcoxon test for SPC.

**Table 5 animals-13-01319-t005:** Health-related parameters of Control (*n* = 112) and Magnesium butyrate cows (*n* = 107) diagnosed within the first 70 DIM.

Item	Control	Magnesium Butyrate	*p*-Value ^1^
Calving assistance rate (%) ^2^	36.9	21.3	0.012
RFM rate (%) ^3^	14.4	3.70	0.008
Embryonic/fetal mortality rate (%)	32.4	13.9	0.001
Prevalence of health disorders (%)	21.6	21.3	1.000
Culling rate within 70 DIM (%)	16.2	13.0	0.568

^1^ Statistical significances between Control and Magnesium butyrate groups are based on results from the Fisher’s exact test. ^2^ Assisted calving was considered as a calving score higher than 1. Calving ease was scored using a 1 to 4 scale following Mee [33]. ^3^ RFM (retained fetal membranes) was defined as the non-expulsion of fetal membranes as assessed at 24 h after calving.

## Data Availability

The data are available on request from the corresponding author.

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
