# Peer review of "Prepartum Magnesium Butyrate Supplementation of Dairy Cows Improves Colostrum Yield, Calving Ease, Fertility, Early Lactation Performance and Neonatal Vitality"

_animals, 2023, doi:10.3390/ani13081319_

Round 1

Reviewer 1 Report

Title: Prepartum Magnesium Butyrate Supplementation of Dairy Cows Improves Colostrum Yield, Calving Ease, Fertility, Early Lactation Performance and Neonatal Vitality

GENERAL CONSIDERATIONS:

The paper focused on the effects of prepartum magnesium butyrate supplementation in dairy cows on colostrum yield, calving ease, fertility, early lactation performance and neonatal vitality. The experiment appears to be had been well-designed, and the methodologies are adequate. However, the manuscript shows several problems that should be improved before the article can be published.

SPECIFIC COMMENTS

Abstract: The sentence below is the same as in Simple Summary (L15-19).

"Butyrate promotes rumen epithelium growth and function, however, the effect of prepartum butyrate supplementation on dairy cow productivity, health and their offspring has  not been extensively studied."

- What was the raw material incorporated into the cows' diet?  "MgB encapsulated in a fat matrix (Rumen-Ready®, Palital Feed Additives B.V., Velddriel, The Netherlands)". However, shouldn't he appear in Table 1 of the diet?

- The introduction is excessively long and unclear, with very exhaustive descriptions of the literature and sometimes inconsistent. These problems do not help to support the hypothesis clearly and understand the objectives of the work.

The statistical design used needs to be described.

Please describe statistical units when making comparisons between treatments in the abstract (P-valeu; SEM, SD ????)

Energy Units (ME or TDN) must be added in Table 1.

Discussion: The lack of a physiological mechanism in this study is clear. The authors only work on a few factors which will not give us a clue for understanding the changes in the body of cows. Since MgB contains can change secondary metabolites which might affect rumen microbiota, having this data for this study is necessary. The rumen microbiota later can change the volatile fatty acids in tissues and other organs from changes of microorganisms.

Author Response

First of all, we would like to thank you for all the extremely pertinent observations, which we followed and allowed us a correction in our manuscript. We believe that the value of our paper has significantly improved.

Reviewer 2 Report

The authors have done a good job that can be published in its entirety.

I just advise not mention the examination of somatic cell count because it is not so important in the colostrum and furthermore the sample has been frozen.

Author Response

We are very grateful for the suggestion, we followed in improving the scientific quality of the manuscript.

Reviewer 3 Report

The manuscript has scientific merit.The study hypothesis is valid and proper research tools were used to test it.The manuscript is well written and follows a logical sequence that allows good understanding by readers.However, some obscure points need clarification by the authors:

1 - The amount of fat supplemented with Mg (~ 45 g/cow/day ~ is that correct?) could affect fiber digestion in the rumen?

2 - Could the amount of fat supplemented with Mg serve as a supplementary energy source and affect the studied response variables that would confuse the results with the effects of Mg supplementation?

3 - What was the significance level adopted to test the random effects used on statistical models?Please describe this in the manuscript.

4 - Please describe better the procedures used to measure BCS (item 2.7)

5 - Please present the quantitative data in all tables with 3 significant digits (example; 7.6 has only 2 significant digits, while 1750.3 does not need .3 and can be presented only as 1750. On the other hand, 2.70 and 18.3, for example, are adequate ways to present data with correct significant digits).

Author Response

Thank you for your constructive comments. We have corrected all the requested points as explained below. All the changes asked by the reviewer appear in tracking mode. Thank you again for your comments and time spent reading our work.

Round 2

Reviewer 1 Report

Most of the questions were answered or corrected. I have just a minor notes.

Table 2: Some data showed strong trends that in my opinion should be discussed (P-value between 0.05<0.1) for Parity. This would improve the discussion.

Table 4: Remove horizontal lines

Author Response

We would like to thank you again for your comments.

Reviewer 3 Report

Authors have covered all of the issues and the manuscript is ready for publication in Animals.

Author Response

many thanks for your positive comment